# Potential Effects of 25-Hydroxycholecalciferol on the Growth Performance, Blood Antioxidant Capacity, Intestinal Barrier Function and Microbiota in Broilers under Lipopolysaccharide Challenge

**DOI:** 10.3390/antiox11112094

**Published:** 2022-10-24

**Authors:** Lianhua Zhang, Jian Wang, Xiangshu Piao

**Affiliations:** 1State Key Laboratory of Animal Nutrition, College of Animal Science and Technology, China Agricultural University, Beijing 100193, China; 2Key Laboratory of Plant Resources and Beijing Botanical Garden, Institute of Botany, Chinese Academy of Sciences, Beijing 100093, China

**Keywords:** 25-hydroxycholecalciferol, blood antioxidant status, intestinal health, microbiota, broilers

## Abstract

Our experiment was to detect the effects of 25-hydroxycholecalciferol (25OHD_3_) on antioxidant capacity, immune status and gut health of broilers under lipopolysaccharide (LPS) challenge. In total, 108 male Arbor Acre broilers (48.5 ± 0.4 g) were allotted to three treatment groups containing six replicates for each group with six birds per replicate: (1) corn-soybean basal diet + injection of sterile saline (CON group); (2) corn-soybean basal diet + an injection of LPS (LPS group); (3) corn-soybean basal diet with 50 μg/kg 25OHD_3_ + injection of LPS (LPS + 25-D group). At the end of the experiment, birds were intraperitoneally injected with LPS in the LPS and LPS + 25-D groups based on the dosage of 5.0 mg/kg BW, or the equivalent volume of 0.9% sterile saline in the CON group. At 4 h postinjection, blood samples, jejunal and ileal tissues and cecal digesta were collected to analyze blood antioxidant capacity, intestinal barrier function and microbiota. The results showed that broilers challenged with LPS had significantly higher BW loss than the CON group, and 25OHD_3_ alleviated BW loss induced by the LPS challenge. 25OHD_3_ alleviated the LPS-induced decline (*p* < 0.05) in serum activities of superoxide dismutase (SOD) and immunoglobulin G (IgG), as well as prevented the LPS-induced increase (*p* < 0.05) in serum content of tumor necrosis factor-α (TNF-α). 25OHD_3_ significantly increased villus height in the jejunum and the relative mRNA abundance of Occludin in the jejunum and ileum, as well as prevented the LPS-induced increase in the jejunal content of interferon-γ (IFN-γ) compared with the LPS group. Compared with the LPS group, 25OHD3 significantly increased *Lactobacillus* abundance and decreased *Lachnoclostridium* abundance in the cecal digesta, as well as had the potential to enhance metabolite contents including propionate, isobutyrate, butyrate and total SCFA. The correlation analysis revealed that BW loss and serum contents of TNF-α, IL-1β and D-lactate were positively correlated with *Lachnoclostridium* and negatively correlated with *Lactobacillus* (*p* < 0.05). Overall, 25OHD_3_ partially improves the antioxidant status, immunity, intestinal barrier and microbial composition of broilers under the LPS challenge.

## 1. Introduction

The consumption of chicken has recently increased worldwide due to its relatively lower fat content, high nutritional value and distinct flavor. However, broilers are exposed to frequent challenges such as pathogenic infection, drug abuse and other external elements in modern intensive production, which could induce oxidative damage and immunological stress [1,2]. Under a stress response, birds are prone to infecting intestinal diseases, which impairs the body’s health and production performance and further leads to a decline in the efficiency of poultry production [3,4]. Lipopolysaccharide (LPS) is one of the components that make up the outer membrane of gram-negative bacteria that could lead to acute systemic inflammation and trigger an imbalance between oxidation and antioxidant defense systems in broilers [2,5]. The injection of LPS in broilers is considered to be a suitable experimental model for investigating systemic inflammation response and oxidative damage in poultry science [5,6]. LPS could cause damage to normal nutrient absorption and mucosal integrity in the small intestine and interfere with the structure and metabolism of cecal microflora in broilers, which could lead to compromised growth performance [7,8]. However, in the European Union and some other countries including China, the use of antibiotic growth promoters has been forbidden in livestock feed. LPS-induced excessive immune response and oxidative damage of broilers could be alleviated by feed additives including vitamins and plant extracts, etc. [9,10]. Obviously, it is important to develop a novel and effective feed additive for regulating the immune status and antioxidant capacity of broilers under stress responses.

In recent years, 25-hydroxycholecalciferol (25OHD_3_) has been becoming more popular based on its commercial production and higher bioavailability. In comparison with vitamin D_3_, adding 25OHD_3_ to diets could avoid a 25-hydroxylation reaction due to the presence of a hydroxyl group, which indicates that exogenous supplementation of 25OHD_3_ is directly used by animals in a ready-to-use active form [11]. Except for the traditional role of maintaining optimal calcium and phosphorus homeostasis, vitamin D has beneficial effects on modulating the body’s immunity, antioxidant status and gut health [12,13,14]. Moreover, 25OHD_3_ could regulate the microbial community in the hindgut of monogastric animals [15]. However, whether 25OHD_3_ exerts beneficial effects on blood antioxidant capacity, immune status, gut barrier function and the microbiota of broilers under the LPS challenge is not fully understood.

It is important to highlight that LPS was used as a model of health challenges and does not necessarily represent a potential problem for broiler chick production like bacterial, fungal, or viral infections released in inflammatory diseases. The principal hypothesis of the study is to show that 25OHD_3_ has antioxidant, immunological and gut health properties when broilers are challenged with LPS. Therefore, this study was conducted to evaluate the effects of 25OHD_3_ on the antioxidant potential, immune function, intestinal barrier and microbiota of broilers under the LPS challenge.

## 2. Materials and Methods

### 2.1. Experimental Animals and Design

A total of 108 one-day-old male broilers (Arbor Acre) with similar body weight (48.5 ± 0.4 g) were allocated to three treatment groups containing six replicates for each group with six birds per replicate: (1) CON group, fed with a basal diet and injected with 0.9% sterile saline; (2) LPS group, fed with a basal diet and challenged with *Escherichia coli* O55:B5-derived LPS (#L2880; Sigma-Aldrich, St. Louis, MO, USA); (3) LPS + 25OHD_3_ group, fed with a basal diet containing 50 μg/kg 25OHD_3_ (0.125%; Haineng Bioengineering Co., Ltd., Rizhao, China) and challenged with LPS. All experimental birds were reared up to 21 days of age in wire-floored cages and provided ad libitum access to fresh water and mash feed. The management procedure of broilers followed the guidelines described by Zhang et al. [16] On day 21, birds were weighed and intraperitoneally injected with LPS in the LPS and LPS + 25-D groups according to the dosage of 5.0 mg/kg BW, or the equivalent volume of 0.9% sterile saline in the CON group. After LPS stimulation, mash feed was withdrawn and the weights of broilers at the start and end of the 4-h LPS challenge were obtained to calculate BW loss. The ingredients and nutrient content of the basal diet are shown in Table 1 according to NRC (1994) [17].

### 2.2. Sample Collection

At 4 h postinjection, blood from six birds per treatment (one bird from each replicate) was obtained from the wing veins and serum samples were separated after centrifugation at 3000× *g* for 10 min at 4 °C. The same birds were sacrificed by injecting 50 mg/kg of pentobarbital sodium. The 2 cm segments of the middle jejunum and ileum were collected, washed gently with 0.9% sterile saline and then immersed in 4% paraformaldehyde. The mucosal samples were scratched gently using a sterile glass slide from the middle jejunum and ileum. The cecal digesta was obtained and then stored at −80 °C.

### 2.3. Serum Parameters

Serum contents of total antioxidant capacity (T-AOC), catalase (CAT), superoxide dismutase (SOD), glutathione peroxidase (GSH-Px), malondialdehyde (MDA), diamine oxidase (DAO) and endotoxin were analyzed by assay kits (Nanjing Jiancheng Bioengineering Institute, Jiangsu, China). The serum D-lactate level was detected by an ELISA kit (Luyuan Byrd Biotechnology Co., Ltd., Beijing, China). Serum contents of inflammatory cytokines such as interleukin (IL)-1β, IL-6, IL-10, tumor necrosis factor-α (TNF-α) and interferon-γ (IFN-γ) were determined by ELISA kits (Nanjing Jiancheng Bioengineering Institute, Jiangsu, China). Serum concentrations of immunoglobulins containing IgA, IgG and IgM were also determined using assay kits from Nanjing Jiancheng Bioengineering Institute (Nanjing, China).

### 2.4. Small Intestinal Morphology Analysis

The morphological structure of the small intestine was analyzed according to the method described by Zhang et al. [18]. Briefly, the intestinal samples were dehydrated, embedded in paraffin wax, and sectioned and stained with eosin and hematoxylin after a 24-h fixation. The villus height and crypt depth were calculated by a calibrated 10-fold eyepiece graticule according to the 10 visual fields of each slice.

### 2.5. The mRNA Abundance of Tight Junction Proteins

Total RNA was isolated from the intestinal mucosa using a Trizol reagent (CWBIO Biotech Co., Beijing, China). The purity and integrity of isolated RNA were detected by a spectrophotometer (Nanodrop ND-1000; Thermo Fisher Scientific, Wilmington, DE, USA). Total RNA was reverse transcribed using the PrimeScript^RT^ Reagent Kit (TaKaRa, Dalian, China). A quantitative real-time PCR was conducted by the ABI Prism 7500 Sequence Detection System (Applied Biosystems, Foster City, CA, USA). The mixture contained 1 μL of cDNA template, 5 μL of SYBR Premix Ex Taq II, 0.4 μL of each forward and reverse primer (10 μM), 0.2 μL of ROX Reference Dye II and 3 μL of sterile water. The reaction conditions were 10 s at 95 °C, 40 cycles of 5 s at 95 °C and 35 s at 60 °C. The relative mRNA expression was determined by the 2^−ΔΔCT^ method and normalized to the reference gene glyceraldehyde-3-phosphate dehydrogenase (*GAPDH*) [19]. Primer sequences were listed in Table 2.

### 2.6. Analysis of Intestinal Mucosal sIgA and Inflammatory Cytokines

The contents of secretory immunoglobulin A (sIgA) and inflammatory cytokines (IL-1β, IL-6, IL-10, TNF-α and IFN-γ) in the mucosal samples of jejunum and ileum were detected by ELISA kits (Nanjing Jiancheng Bioengineering Institute, Jiangsu, China).

### 2.7. Microbial Community

Total genomic DNA of cecal contents was extracted by a Stool DNA Kit (Omega Bio-TEK, Norcross, GA, USA) according to the manufacturer’s procedure. The quantity of DNA was analyzed by a NanoDrop 2000 UV-vis spectrophotometer (Thermo Scientific, Wilmington, DE, USA) and the integrity of DNA was checked by 1% agarose gel electrophoresis. The V3-V4 hypervariable region of the 16S rRNA gene was amplified via the universal primers 338F (5′-barcode-ACTCCTRCGGGAGGCAGCAG-3′) and 806R (5′-GGACTACCVGGGTATCTAAT-3′). Amplicons were extracted from 2% agarose gels, purified using the AxyPrep DNA Gel Extraction Kit (Axygen Biosciences, Union City, CA, USA) and quantified with a QuantiFluor TM-ST fluorometer (Promega, Madison, WI, USA). Purified amplicons were pooled and then paired-end sequenced using the Illumina MiSeq platform (Illumina, San Diego, CA, USA). Demultiplexing and quality filtering of raw sequences were performed by QIIME (version 1.17). Operational taxonomic units (OTU) were clustered using UPARSE software according to a similarity of more than 97%. The taxonomy of 16S rRNA gene sequences was determined by the RDP Classifier (http://rdp.cme.msu.edu/ accessed on 10 November 2021) with a confidence threshold of 70%. Data analysis and processing were conducted based on the Majorbio cloud platform (Majorbio Bio-pharm Technology Co., Ltd., Shanghai, China).

### 2.8. Microbial Metabolites

The composition of short-chain fatty acids (SCFA) in the cecal digesta was detected based on the method described by Zhang et al. [20]. Briefly, digesta samples (0.5 g) were mixed with 8 mL of distilled water, shaken and then centrifuged at 3000 × g for 5 min. The collected supernatants were diluted 50 times with distilled water. After filtration through a 0.22 μm membrane, each sample was determined by an Ion Chromatography system (DIONEX ICS-3000; Thermo Fisher, Waltham, MA, USA). The concentrations of SCFA are presented as μg/g of the cecal digesta.

### 2.9. Statistical Aanalysis

Data analysis was performed using One-Way ANOVA via the GLM procedure of SAS 9.4 (SAS Inst. Inc., Cary, NC, USA). Statistical differences among the three treatments were detected by Student-Neuman-Keul’s multiple range tests. The relative abundance of microbial composition in the cecal digesta was determined by the Kruskal-Wallis test. Correlations between microbial composition at the genus level, BW loss and serum parameters were evaluated by Pearson’s correlation test. *p* < 0.05 was defined as significant difference, whereas 0.05 ≤ *p* < 0.10 was defined as tendency.

## 3. Results

### 3.1. Growth Performance before the LPS Challenge and BW Loss after the LPS Challenge

No difference was detected in performance among the three groups before the LPS challenge (Appendix A). The ratio of live BW loss after LPS challenge to BW before inoculation is shown in Figure 1. LPS enhanced (*p* < 0.05) BW loss of broilers compared to the CON group, and 25OHD_3_ alleviated (*p* < 0.05) BW loss of broilers c subjected to LPS.

### 3.2. Serum Oxidative Status

Compared to the CON group, the LPS challenge reduced (*p* < 0.05) the activities of T-AOC, CAT and SOD (Table 3). Dietary 25OHD_3_ supplementation prevented the LPS-induced decrease (*p* < 0.05) in the serum SOD activity.

### 3.3. Serum Immunological Parameters

Compared to the CON group, broilers injected with LPS had higher (*p* < 0.05) contents of D-lactate, TNF-α and IL-1β (Table 4). However, dietary 25OHD_3_ supplementation prevented the LPS-induced increase (*p* < 0.05) in the serum TNF-α content. Compared to the CON group, LPS stimulation reduced (*p* < 0.05) serum IgG content, and 25OHD_3_ alleviated (*p* < 0.05) decrease in serum IgG content of broilers caused by LPS challenge.

### 3.4. Intestinal Morphology

Compared to the CON group, broilers injected with LPS had significantly lower villus height in the jejunum (Table 5). However, 25OHD_3_ markedly improved villus height in the jejunum compared to the LPS group. There were no significant differences in ileal morphology among the three treatments.

### 3.5. Tight Junction Gene Expression

Broilers injected with LPS had lower (*p* < 0.05) mRNA expression of Occludin in the jejunum and ileum than in the CON group (Table 6). However, 25OHD_3_ enhanced (*p* < 0.05) the mRNA expression of Occludin in the jejunum and ileum in comparison with the LPS group.

### 3.6. Inflammatory Cytokines and sIgA in the Intestinal Mucosa

LPS stimulation enhanced (*p* < 0.05) jejunal TNF-α and IFN-γ contents and ileal TNF-α content compared with the CON group (Table 7). However, dietary 25OHD_3_ supplementation significantly prevented the LPS-induced increase in the jejunal IFN-γ content. No significant change was observed in mucosal sIgA level among the three treatments.

### 3.7. Cecal Microbiota

The diversity of cecal microbiota is shown in Figure 2 and Figure 3. The Venn analysis identified 28, 19 and 16 unique OTU in the CON, LPS and LPS + 25-D groups, respectively (Figure 2A). In comparison with the CON group, the LPS challenge reduced (*p* < 0.05) Shannon index and tended to enhance (*p* = 0.096) Simpson index in the cecal digesta (Figure 2B). However, no significant differences were discovered for the α-diversity indices between CON and LPS + 25-D groups. 

The principal component analysis demonstrated that greater changes were observed in the cecal digesta among the three treatments (Figure 3A). The UPGMA tree analysis indicated that there were obvious differences in the structure of cecal bacteria among the three treatments (Figure 3B).

The composition of cecal microbiota is shown in Figure 4. At the phylum level, the dominant bacteria were Firmicutes and Proteobacteria, accounting for more than 98% (Figure 4A). At the genus level, the predominant bacteria consisted of *unclassified_f_Lachnospiraceae*, *Ruminococcus_torques_group*, *Faecalibacterium*, *Lactobacillus*, *Blautia*, *norank_f_norank_o_Clostridia_UCG-014*, *norank_f_norank_o_Clostridia_vadinBB60_group*, *Escherichia-Shigella*, *Butyricicoccus*, *Sellimonas*, *Romboutsia*, *UCG-005*, *norank_f_Ruminococcaceae*, *Subdoligranulum*, *Erysipelatoclostridium*, *Lachnoclostridium*, *norank_f_norank_o_RF39* and *norank_f_Oscillospiraceae* (Figure 4B). Compared with the CON group, LPS challenge tended to decrease Firmicutes abundance and increase Proteobacteria abundance in the cecal digesta (Figure 4C). For the predominant bacteria at the genus level, broilers challenged with LPS had significantly lower *Lactobacillus* abundance and higher *Lachnoclostridium* abundance in the cecal digesta compared with the CON group (Figure 4D). However, 25OHD_3_ enhanced (*p* < 0.05) *Lactobacillus* abundance and reduced (*p* < 0.05) *Lachnoclostridium* abundance in the cecal digesta compared with the LPS group. Furthermore, *Blautia* abundance in the cecal digesta of LPS + 25-D was higher (*p* < 0.05) in comparison with the LPS group, and *Sellimonas* abundance in the cecal digesta of LPS and LPS + 25-D groups was higher (*p* < 0.05) in comparison with the CON group.

### 3.8. Fermentation Metabolites of Cecal Digesta

Compared to the CON group, the LPS challenge reduced (*p* < 0.05) the contents of propionate, isobutyrate, butyrate and total SCFA in the cecal digesta (Table 8). However, dietary 25OHD_3_ supplementation had the potential to alleviate the LPS-induced decrease in the contents of propionate, isobutyrate, butyrate and total SCFA.

### 3.9. Association between Cecal Microbiota, BW Loss and Serum Immunological Parameters

The results of the correlation analysis between cecal microbiota, BW loss and serum immunological parameters are shown in Figure 5. For differential bacteria at the genus level, BW loss and serum contents of IL-1β, TNF-α and D-lactate were positively associated with cecal *Lachnoclostridium* and negatively associated with cecal *Lactobacillus* (*p* < 0.05). Serum IgG level was negatively associated with cecal *Lachnoclostridium* (*p* < 0.05). Moreover, serum IL-6 content was positively associated with cecal *Sellimonas* (*p* < 0.05). No significant correlations were observed between cecal *Blautia* and serum immunological parameters.

## 4. Discussion

Live BW loss due to enhanced catabolism and decreased anabolism under stress response is of economic importance in poultry production and has been considered a predictable indicator of stress in broilers [21,22]. Recent studies have demonstrated that 25OHD_3_ could improve the performance of broilers [16,23]. Our study showed that 25OHD_3_ supplementation decreased the BW loss of broilers compared with the LPS group, indicating that 25OHD_3_ had a stronger positive effect on the improvement of performance in broilers under LPS stimulation. The findings of this present study substantiate the results that 25OHD_3_ supplementation alleviated the compromised performance of broilers during an experimental LPS injection [9]. The changes in BW loss among the three treatments may be related to the changes in antioxidant status, immunity, intestinal barrier function and the microbial composition of broilers. Therefore, we further determined the effects of adding 25OHD_3_ to diets on antioxidant potential, immune status and intestinal function of broilers under the LPS challenge to explore the underlying mechanisms.

In the poultry industry, broilers inevitably face the oxidative stress associated with bacteria or their products such as LPS, which could impair health status [2]. Oxidative stress refers to the unbalanced status between the generation rate of reactive oxygen species (ROS) and their clearing by the body’s antioxidant system [24]. Serum antioxidant status could indicate the host’s resistibility to oxidative damage, and a higher degree of antioxidant potential contributes to relieving oxidative stress [25]. Wu et al. [26] reported that LPS disturbed the balance status between oxidant and antioxidant systems, which led to oxidative damage to broilers. The previous study showed that 25OHD_3_ could maintain the delicate balance between the oxidant and antioxidant systems of broilers by modulating the nonenzymatic antioxidant defense system reflected by T-AOC and serum activities of antioxidant enzymes including CAT and GSH-Px [16]. This study showed that the LPS challenge significantly led to oxidative stress in broilers due to a decrease in serum activities of T-AOC, CAT and SOD, whereas, dietary 25OHD_3_ supplementation significantly alleviated the LPS-induced decrease in the serum activity of SOD. Based on this result, we concluded that 25OHD_3_ alleviates LPS-induced oxidative stress of broilers by stimulating the non-enzymatic and enzymatic antioxidant defensive systems.

The morphological structure is a major indicator to assess the absorption ability and mucosal integrity of the small intestine [18]. Morphological changes in the small intestine, such as villus atrophy and crypt hyperplasia, indicate nutrient malabsorption and growth retardation in animals [27]. Microbial challenges and their products such as LPS could induce marked changes in the morphological structure including shorter villus height and a lower ratio of villus height to crypt depth in the small intestine [2,28], and these changes, in turn, lead to growth retardation in broilers. The present study showed that 25OHD_3_ supplementation significantly alleviated the LPS-induced decrease in villus height of the jejunum. Higher villus height caused by 25OHD_3_ supplementation reflected the greater ability of nutrient digestion and absorption, thereby increasing performance in broilers. It has been acknowledged that putrescine has a certain modulatory effect on the potential mode of vitamin D regulating intestinal morphology and function [29]. 1,25-dihydroxycholecalciferol [1,25(OH)_2_D_3_] is a pleiotropic steroid hormone that could improve the activities of spermidine N-acetyltransferase and ornithine decarboxylase used to catalyze the production of putrescine [30,31]. One possibility is that an increased 25OHD_3_ level in serum may result in a higher 1,25(OH)_2_D_3_ level, and the other possible mode is that 25OHD_3_ may act directly on vitamin D receptor (VDR) in a VDR independent action of 1,25(OH)_2_D_3_ [32], although its affinity for VDR is less than 1,25(OH)_2_D_3_. However, the exact mechanism behind increased intestinal morphology caused by 25OHD_3_ supplementation needs further research.

The intestinal epithelial barrier consists of epithelial cells and intercellular multiprotein complexes including ZO-1, occludin and claudin, which could prevent pathogens, antigens and toxins from passing through the intestinal lumen into the circulating system [33]. Tight junction proteins are critical for regulating intestinal permeability and maintaining the integrity of the intestinal tissue. Moreover, DAO, D-lactate and endotoxin are considered as important indicators related to intestinal permeability, and their serum concentrations could indicate the intestinal barrier integrity [34]. Chen et al. [28] reported that the intraperitoneal LPS challenge impaired the intestinal barrier of broilers based on increased serum D-lactate levels and lower mRNA abundances of intestinal tight junction proteins. Our study showed that 25OHD_3_ had the potential to alleviate the LPS-induced increase in serum D-lactate level, resulting in lower intestinal permeability and improved intestinal barrier in broilers. The present results also showed that the inclusion of 25OHD_3_ markedly promoted Occludin mRNA expression in the mucosa of the jejunum and ileum compared with LPS group, which, once again, indicated an improved intestinal barrier. Overall, adding 25OHD_3_ to diets could relieve intestinal barrier injury induced by the LPS challenge. Compared with vitamin D_3_, 25OHD_3_ could enhance 1α-hydroxylase activity in broilers. 1,25(OH)_2_D_3_ has been shown to protect the intestinal barrier from damage caused by the LPS challenge [35]. The protective effects of 1,25(OH)_2_D_3_ on the intestinal barrier are mediated by VDR, and the receptor might play a vital role in the main pathway which modulates all of the factors including inflammation and microbiota composition [35,36].

The inflammatory cytokines have important effects on regulating intestinal immune response and are taken as important mediators for preventing or being susceptible to infection and certain intestinal disorders. The present study showed that the LPS challenge enhanced serum contents of TNF-α and IL-1β in comparison with the CON group. Our results from intestinal mucosa also showed that broilers in the LPS group had higher concentrations of TNF-α and IFN-γ. Our findings were similar to the previous results of Chen et al. [28] that LPS stimulation enhanced the concentrations of pro-inflammatory cytokines in serum and the intestinal mucosa of broilers. The pro-inflammatory cytokines have been shown to impair the intestinal tight junction and disrupt the intestinal barrier [37,38]. Therefore, lower contents of pro-inflammatory cytokines in the LPS + 25-D group may be part of the reason for improving the intestinal barrier of broilers. In addition, our findings suggested that 25OHD_3_ markedly enhanced the serum IgG level in broilers compared to the LPS group. Serum immunoglobulin levels reflect the immune function of animals to a certain extent, which could help relieve immunological stress, promote health status and improve growth performance [39,40]. In animals, 1α-hydroxylase has been identified in T-cells, B-cells, macrophages and the small intestine [41,42,43,44]. It was reported that immune cells and the small intestine may show a greater response to this secondary metabolite of vitamin D, 25OHD_3_ [45], which partly explained the improved humoral immunity of broilers. Overall, these results suggested that 25OHD_3_ could promote the immunity of broilers after LPS stimulation by changing the production of inflammatory cytokines and immunoglobulins.

The cecal microbiota regulates nutritional metabolism and immune function, such as nitrogen cycling, providing their host with vitamins, amino acids and SCFA, and competitively preventing pathogenic infections [46]. The microbial community in the cecum of broilers could be affected by several factors, such as diet compositions, feed additives, raising conditions and pathogenic microorganisms [47]. In particular, the LPS challenge could result in significant changes in the cecal microbiota of broilers [4,7]. An adequate vitamin D level is considered as an essential factor for improving microbial composition in the intestine [48]. The α-diversity is taken as the diversity of the intestinal microbiota within one sample, including species diversity (Shannon and Simpson) and species richness (Chao and Ace) [15], and high α-diversity helps maintain stable gut ecosystems that inhibit pathogen proliferation [49]. Our study showed that 25OHD_3_ alleviated the LPS-induced decrease in the Shannon index of cecal digesta, revealing that 25OHD_3_ could improve intestinal immunity and barrier function in broilers under the LPS challenge. The β-diversity shown by the PCA and UPGMA tree also suggested that cecal bacteria responded differently to different experimental groups. At the phylum level, the main bacteria in the cecal digesta were Firmicutes and Proteobacteria, which were consistent with the previous results described by Lucke et al. [7]. Firmicutes are closely related to energy metabolism and anti-inflammatory response, and a lower abundance of Firmicutes has a relationship with the occurrence of intestinal inflammation [50,51]. Many pathogens, including *Salmonella*, *Escherichia*, *Helicobacter* and *Pseudomonas*, belong to the Proteobacteria phylum [52]. Our findings demonstrated that the LPS challenge tended to decrease Firmicutes abundance and increase Proteobacteria abundance in the cecal digesta, indicating that the LPS challenge disrupted the intestinal microbial community and impaired the intestinal homeostasis of broilers. Down to the genus level, 25OHD_3_ supplementation enhanced *Lactobacillus* abundance and decreased *Lachnoclostridium* abundance in the cecal digesta compared with the LPS group. *Lactobacillus* is regarded as probiotic bacteria, which plays important role in inhibiting inflammatory response by mediating the production of cytokines, improving intestinal barrier function by stimulating the tight junction integrity and combating infection by pathogens such as *Salmonella* [53,54]. In this study, the correlation analysis showed that BW loss and serum contents of D-lactate, IL-1β and TNF-α were negatively correlated with cecal *Lactobacillus* abundance, confirming the beneficial effects of *Lactobacillus* on gut barrier function and performance. Wu et al. [55] demonstrated that the *Lactobacillus* species, including *L. rhamnosus* and *L. plantarum*, increased the expression and activity of VDR in the intestinal epithelial cells, and *Lactobacillus* was depleted in the feces of VDR^–/–^ mice. Therefore, higher *Lactobacillus* abundance may be caused by the vitamin D/VDR signaling pathway, but the underlying mechanism needs further research. *Lachnoclostridium* has been reported to act as a pathogenic source of inflammatory disease [56], which could disrupt intestinal health and impair performance in animals. The correlation analysis also showed that BW loss and serum contents of TNF-α, IL-1β and D-lactate were positively correlated with cecal *Lachnoclostridium* abundance, highlighting the negative effects of *Lachnoclostridium* on gut barrier function and performance. Tangestani et al. [48] suggested that vitamin D may connect to some genera of the Lachnospiaceae family such as *Blautia*. The beneficial genus *Blautia* has been shown to produce SCFA, and it is positively correlated with metabolic homeostasis [57]. Our findings suggested that adding 25OHD_3_ to diets markedly promoted the relative abundance of *Blautia* compared with the CON and LPS groups, and no clear difference was discovered between the CON and LPS groups. These results indicated that the changes in *Blautia* abundance were not related to the LPS challenge and were only associated with 25OHD_3_ supplementation. SCFA, including acetate, propionate and butyrate, have beneficial effects on alleviating intestinal inflammation and preventing the imbalance of the intestinal flora. SCFA, especially butyrate, could stimulate the growth of beneficial bacteria including Lactobacillus in poultry, which in turn promotes an increase in SCFA, improves the intestinal barrier function and inhibits the proliferation of pathogenic bacteria including *Salmonella* and *E. coli* [58,59]. The previous study also suggested that propionate and butyrate suppressed the LPS-induced release of TNF-α and inhibited the downstream pathway of NF-κB signaling [60]. The present study showed that 25OHD_3_ had the potential to prevent the LPS-induced decrease in propionate, isobutyrate, butyrate and total SCFA, which may be partly responsible for the improved immune responses and enhanced barrier function in broilers under stress response. Overall, 25OHD_3_ could alleviate LPS-induced intestinal injury via modulating the intestinal bacterial compositions and metabolites of broilers.

## 5. Conclusions

Dietary supplementation with 50 μg/kg 25OHD_3_ improved antioxidant capacity, intestinal barrier function and microbiota composition in broilers during an experimental LPS injection by increasing serum contents of SOD and IgG and mRNA expression of jejunal and ileal Occludin, reducing serum TNF-α levels and jejunal IFN-γ content, as well as enhancing cecal Lactobacillus abundance and metabolite contents including propionate, butyrate and total SCFA. Overall, 25OHD_3_ can be used as an effective approach to alleviating BW loss via partially promoting antioxidant capacity, inhibiting inflammatory response, and modulating the gut microbiota and metabolites of broilers under LPS stress.

## Figures and Tables

**Figure 1 antioxidants-11-02094-f001:**
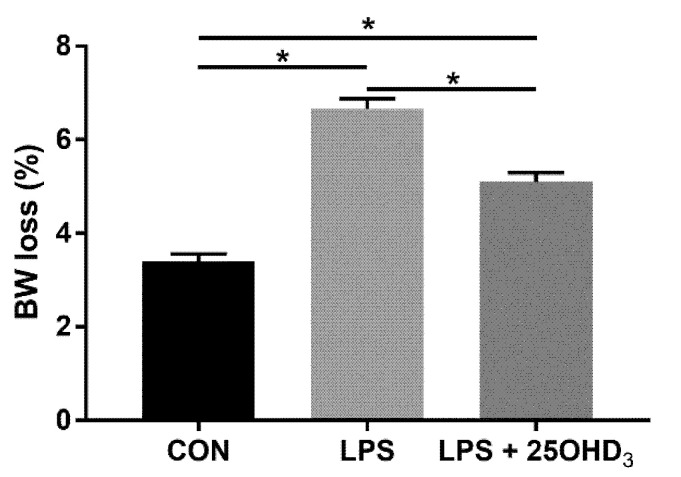
Effects of 25OHD_3_ on body weight loss of LPS-challenged broilers. CON, control; LPS, lipopolysaccharide; LPS + 25OHD_3_, lipopolysaccharide + 25-hydroxycholecalciferol. Values are means with their SEM (*n* = 6). * *p* < 0.05.

**Figure 2 antioxidants-11-02094-f002:**
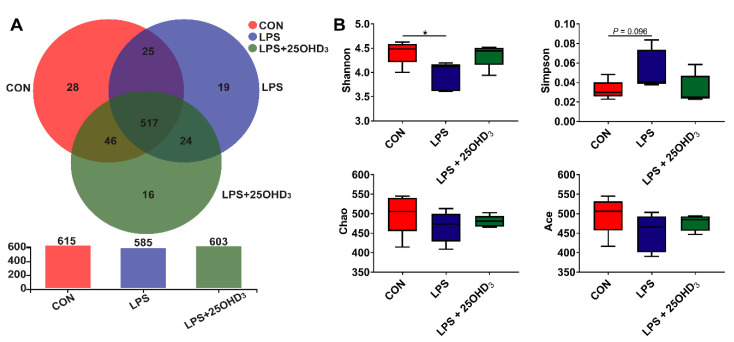
The richness and diversity of cecal microbiota. (**A**) OUT Venn. (**B**) Comparison of α-diversity indices. CON, control; LPS, lipopolysaccharide; LPS + 25OHD_3_, lipopolysaccharide + 25-hydroxycholecalciferol. *n* = 5. * *p* < 0.05.

**Figure 3 antioxidants-11-02094-f003:**
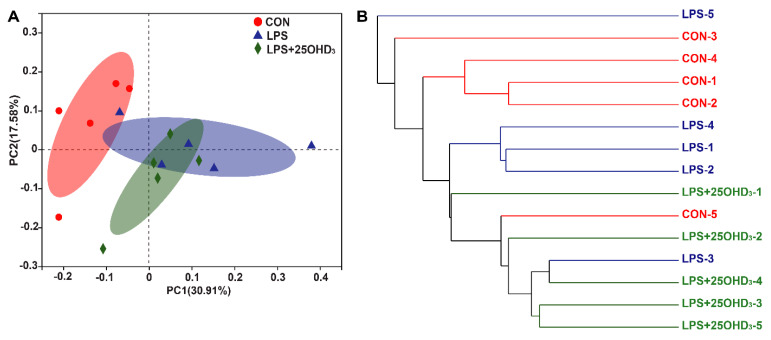
The β-diversity of cecal microbiota structure based on the OUT level. (**A**) Principal component analysis. (**B**) UPGMA tree analysis. CON, control; LPS, lipopolysaccharide; LPS + 25OHD_3_, lipopolysaccharide + 25-hydroxycholecalciferol. *n* = 5.

**Figure 4 antioxidants-11-02094-f004:**
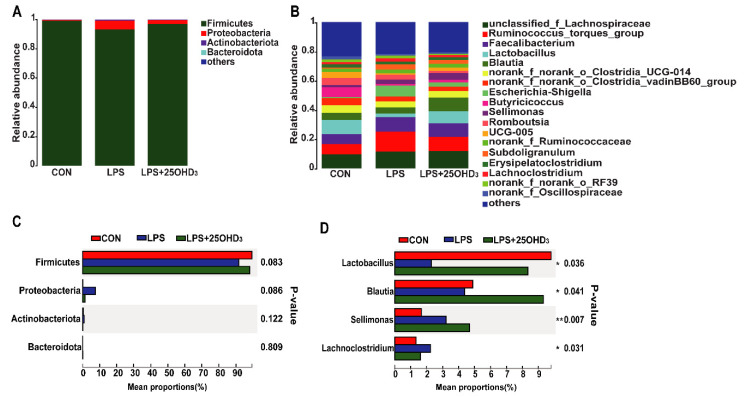
Effects of 25OHD_3_ on cecal microbiota composition of LPS-challenged broilers. (**A**,**B**) Microbiota composition at the phylum and genus levels. (**C**,**D**) Changes in microbiota composition at the phylum and genus levels. CON, control; LPS, lipopolysaccharide; LPS + 25OHD_3_, lipopolysaccharide + 25-hydroxycholecalciferol. *n* = 5. * *p* < 0.05, ** *p* < 0.01.

**Figure 5 antioxidants-11-02094-f005:**
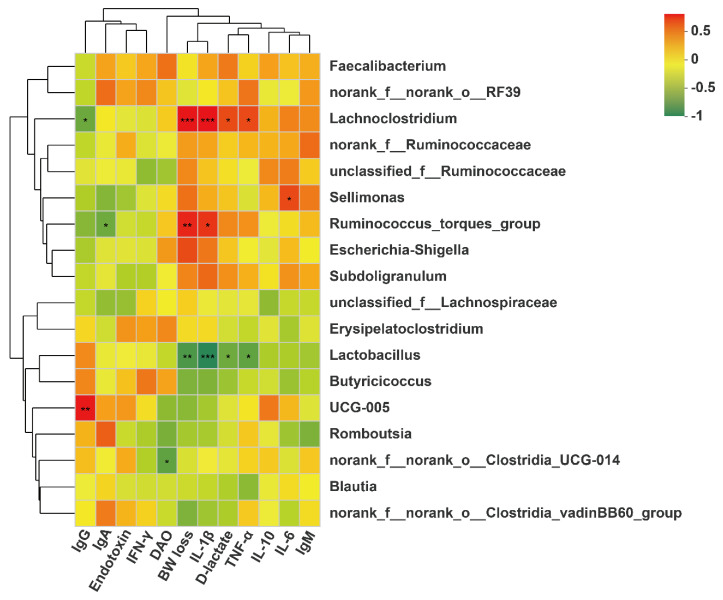
Heatmap of Pearson’s correlation between cecal microbiota (at the genus level), BW loss and serum parameters. ^*^ *p* < 0.05, ^**^ *p* < 0.01, ^***^ *p* < 0.001.

**Table 1 antioxidants-11-02094-t001:** Ingredients and chemical content of the basal diet (as-fed basis, %).

Item	
Ingredients	
Corn	58.65
Soybean meal	30.39
Fish meal	2.00
Corn gluten meal	2.00
Soybean oil	3.20
Limestone	1.30
Dicalcium phosphate	1.50
Sodium chloride	0.30
L-Lysine hydrochloride	0.01
DL-Methionine	0.14
L-Threonine	0.01
Premix ^1^	0.50
Nutrient levels ^2^	
Metabolizable energy, kcal/kg	3050
Crude protein	21.01
Lysine	1.10
Methionine	0.50
Calcium	1.00
Available phosphorus	0.45

^1^ Provided the following per kilogram of diet: 10,000 IU vitamin A; 2760 IU vitamin D_3_; 30 IU vitamin E; 1.2 mg vitamin B_12_; 2 mg vitamin K_3_; 6 mg riboflavin; 40 mg nicotinic acid; 12 mg pantothenic acid; 3 mg pyridoxine; 0.2 mg biotin; 800 mg choline chloride; 10 mg copper; 60 mg zinc; 80 mg manganese; 100 mg iron; 0.3 mg selenium; 0.35 mg iodine.^2^ Calculated nutrient levels.

**Table 2 antioxidants-11-02094-t002:** Primers used in the real-time PCR.

Genes	Primer Sequences (5′ to 3′)	Size, bp	Accession No.
*ZO-1*	F: TCAAGGTCTGCCGAGACAAC	140	XM_003353439.2
	R: ATCACAGTGTGGTAAGCGCA		
Occludin	F: CAGGTGCACCCTCCAGATTG	111	NM_001163647.2
	R: TGGACTTTCAAGAGGCCTGG		
Claudin-1	F: ACAGGAGGGAAGCCATTTTCA	82	NM_001244539.1
	R: TTTAAGGACCGCCCTCTCCC		
*GAPDH*	F: TCGGAGTGAACGGATTTGGC	189	NM_001206359.1
	R: TGACAAGCTTCCCGTTCTCC		

*ZO-1*, zonula occludens-1; *GAPDH*, glyceraldehyde-3-phosphate dehydrogenase.

**Table 3 antioxidants-11-02094-t003:** Effects of 25OHD_3_ on serum antioxidant indices of broilers under LPS stimulation at 4 h postinjection ^1^.

Item	CON	LPS	LPS + 25OHD_3_	SEM	*p*-Value
T-AOC, U/mL	13.97 ^a^	9.54 ^b^	9.82 ^b^	0.67	<0.001
CAT, U/mL	7.20 ^a^	5.19 ^b^	6.38 ^ab^	0.44	0.021
SOD, U/mL	53.97 ^a^	46.06 ^b^	50.83 ^a^	1.11	<0.001
GSH-Px, nmol/mL	13.29	12.68	12.45	0.61	0.619
MDA, nmol/mL	1.21	1.14	1.20	0.08	0.798

CON, control; LPS, lipopolysaccharide; LPS + 25OHD_3_, lipopolysaccharide + 25-hydroxycholecalciferol; T-AOC, total antioxidant capacity; CAT, catalase; SOD, superoxide dismutase; GSH-Px, glutathione peroxidase; MDA, malondialdehyde. ^a,b^ Within a row, values with different superscripts mean significance (*p* < 0.05). ^1^ Means with their SEM (*n* = 6).

**Table 4 antioxidants-11-02094-t004:** Effects of 25OHD_3_ on serum immunological parameters of broilers under LPS stimulation at 4 h postinjection ^1^.

Item	CON	LPS	LPS + 25OHD_3_	SEM	*p*-Value
DAO, U/mL	2.23	2.27	2.27	0.14	0.974
D-lactate, μmol/L	5.37 ^b^	6.31 ^a^	5.99 ^ab^	0.18	0.008
Endotoxin, EU/mL	2.58	2.45	2.45	0.11	0.660
TNF-α, ng/L	66.41 ^b^	76.79 ^a^	63.21 ^b^	2.65	0.007
IFN-γ, pg/mL	7.41	7.51	7.42	0.28	0.963
IL-1β, pg/mL	58.54 ^b^	71.49 ^a^	64.71 ^ab^	1.92	0.001
IL-6, ng/L	19.02	20.31	22.71	1.19	0.116
IL-10, pg/mL	8.06	7.92	8.55	0.37	0.467
IgA, μg/mL	12.32	11.83	11.38	0.49	0.422
IgG, ng/L	9.45 ^a^	6.66 ^b^	8.42 ^a^	0.45	0.002
IgM, μg/mL	2.54	2.54	2.71	0.13	0.587

CON, control; LPS, lipopolysaccharide; LPS + 25OHD_3_, lipopolysaccharide + 25-hydroxycholecalciferol; DAO, diamine oxidase; TNF-α, tumor necrosis factor-α; IFN-γ, interferon-γ; IL-1β, interleukin-1β; IL-6, interleukin-6; IL-10, interleukin-10; IgA, immunoglobulin A; IgG, immunoglobulin G; IgM, immunoglobulin M. ^a,b^ Within a row, values with different superscripts mean significance (*p* < 0.05). ^1^ Means with their SEM (*n* = 6).

**Table 5 antioxidants-11-02094-t005:** Effects of 25OHD_3_ on intestinal morphology of broilers under LPS stimulation at 4 h postinjection ^1^.

Item	CON	LPS	LPS + 25OHD_3_	SEM	*p*-Value
Jejunum					
Villus height, μm	527.50 ^a^	460.58 ^b^	514.94 ^a^	14.17	0.010
Crypt depth, μm	98.26	100.56	101.44	5.29	0.908
Villus height/crypt depth	5.41	4.69	5.10	0.25	0.159
Ileum					
Villus height, μm	447.83	404.11	439.60	13.56	0.083
Crypt depth, μm	83.09	81.06	81.77	4.37	0.946
Villus height/crypt depth	5.46	5.05	5.44	0.29	0.552

CON, control; LPS, lipopolysaccharide; LPS + 25OHD_3_, lipopolysaccharide + 25-hydroxycholecalciferol. ^a,b^ Within a row, values with different superscripts mean significance (*p* < 0.05). ^1^ Means with their SEM (*n* = 6).

**Table 6 antioxidants-11-02094-t006:** Effects of 25OHD_3_ on relative mRNA abundance of intestinal tight junction proteins of broilers under LPS stimulation at 4 h postinjection ^1^.

Item	CON	LPS	LPS + 25OHD_3_	SEM	*p*-Value
Jejunum					
*ZO-1*	1.00	0.93	1.23	0.14	0.302
Occludin	1.00 ^a^	0.61 ^b^	1.10 ^a^	0.09	0.005
Claudin-1	1.00	1.02	1.15	0.10	0.490
Ileum					
*ZO-1*	1.00	0.83	1.18	0.13	0.212
Occludin	1.00 ^a^	0.77 ^b^	1.01 ^a^	0.04	0.003
Claudin-1	1.00	1.09	1.05	0.13	0.872

CON, control; LPS, lipopolysaccharide; LPS + 25OHD_3_, lipopolysaccharide + 25-hydroxycholecalciferol; *ZO-1* = zonula occludens-1. ^a,b^ Within a row, values with different superscripts mean significant (*p* < 0.05). ^1^ Means with their SEM (*n* = 6).

**Table 7 antioxidants-11-02094-t007:** Effects of 25OHD_3_ on sIgA and inflammatory cytokines in the intestinal mucosa of broilers under LPS stimulation at 4 h postinjection ^1^.

Item	CON	LPS	LPS + 25OHD_3_	SEM	*p*-Value
Jejunum					
sIgA, ng/mg	193.15	185.76	181.33	14.12	0.838
TNF-α, ng/g	51.56 ^b^	62.53 ^a^	55.81 ^ab^	3.03	0.016
IFN-γ, pg/mg	5.04 ^b^	6.61 ^a^	4.96 ^b^	0.46	0.040
IL-1β, pg/mg	38.86	49.66	36.70	4.36	0.113
IL-6, ng/g	16.97	20.84	15.60	1.79	0.135
IL-10, pg/mg	7.61	7.76	7.30	0.61	0.859
Ileum					
sIgA, ng/mg	196.36	173.98	181.57	17.11	0.650
TNF-α, ng/g	79.08 ^b^	93.49 ^a^	83.72 ^ab^	3.61	0.037
IFN-γ, pg/mg	8.38	8.33	8.18	0.85	0.987
IL-1β, pg/mg	60.25	58.47	67.15	3.43	0.201
IL-6, ng/g	23.53	25.84	25.85	4.36	0.911
IL-10, pg/mg	10.51	9.30	9.60	1.02	0.693

CON, control; LPS, lipopolysaccharide; LPS + 25OHD_3_, lipopolysaccharide + 25-hydroxycholecalciferol; sIgA, secretory immunoglobulin A; TNF-α, tumor necrosis factor-α; IFN-γ, interferon-γ; IL-1β, interleukin-1β; IL-6, interleukin-6; IL-10, interleukin-10. ^a,b^ Within a row, values with different superscripts mean significance (*p* < 0.05). ^1^ Means with their SEM (*n* = 6).

**Table 8 antioxidants-11-02094-t008:** Effects of 25OHD_3_ on cecal SCFA contents (μg/g) of broilers under LPS stimulation at 4 h postinjection ^1^.

Item	CON	LPS	LPS + 25OHD_3_	SEM	*p*-Value
Acetate	3683	3043	3512	281.56	0.280
Propionate	364.14 ^a^	150.79 ^b^	293.19 ^ab^	51.40	0.030
Isobutyrate	53.33 ^a^	23.98 ^b^	34.31 ^ab^	5.96	0.011
Butyrate	702.53 ^a^	433.12 ^b^	574.56 ^ab^	55.82	0.013
Isovalerate	23.28	13.16	20.64	5.41	0.412
Valerate	31.28	15.47	25.86	4.28	0.056
Total SCFA	4857 ^a^	3679 ^b^	4461 ^ab^	304.57	0.043

CON, control; LPS, lipopolysaccharide; LPS + 25OHD_3_, lipopolysaccharide + 25-hydroxycholecalciferol. ^a,b^ Within a row, values with different superscripts mean significance (*p* < 0.05). ^1^ Means with their SEM (*n* = 6).

## Data Availability

All of the data is contained within the article and Appendix A.

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
