# Peer review of "Potential Effects of 25-Hydroxycholecalciferol on the Growth Performance, Blood Antioxidant Capacity, Intestinal Barrier Function and Microbiota in Broilers under Lipopolysaccharide Challenge"

_antioxidants, 2022, doi:10.3390/antiox11112094_

Round 1
Reviewer 1 Report
Title: Potential Effects of 25-Hydroxycholecalciferol on the Growth Performance, Blood Antioxidant Capacity, Intestinal Barrier Function and Microbiota in Broilers under Lipopolysaccharide Challenge
1.1. Comments to authors:
2. Specific comments:
2.1. Abstract section
I recommended briefly describing the protocol of LPS challenge and response parameters collected before the results paragraph. That information is essential for a quick review of potential readers and helps to best understand the whole paper.
2.2. Introduction section
In the first paragraph, the authors justify that the “broilers are frequently challenging” to many factors mentioned in the environmental pollution. Maybe to be more accurate with this proposition, “environmental pollution” could be changed by “climatic changes” and “global warming” as a direct stressor (principally as heat stress) that induces oxidative damage. Using the term “environmental pollution” involves many factors that not necessarily induce a direct stress response in the animals.
The last paragraph of the introduction section is mentioned that: "Several studies demonstrated that adding 25OHD3 to the diet improved antioxidant status, immune function…", and in the subsequent paragraph: “However, fewer studies were conducted to explore the effect of 25OHD3… under LPS challenge”. Both prepositions are a little contradictory since the authors justify the lack of studies because the 25OHD3 benefits were not explored using the LPS model. Also, the principal hypothesis of the study is to show that 25OHD3 has antioxidant, anti-inflammatory, immunological, and gut health properties. It is important to highlight that LPS was used as a model of health challenge (as a tool) and does not necessarily represent a potential problem for broiler chick production like bacterial, fungal, or viral infections released in inflammatory diseases.
2.3. Materials and methods section
The term “ad libitum” is repeated. Please, edit this paragraph.
2.4. Results section
On item 3.1., I recommend expressing the results of body weight variation in the proportion of initial body weight (before inoculation) or declaring the average of initial body weight between treatments to elucidate that there is no difference at 21-d-old as a result of the treatments (specifically to 25OHD3 supplementation).
On item 3.2. the authors mentioned that “25OHD3 supplementation prevented the LPS-induced decrease in the activities of CAT and SOD”. Also, this affirmation is valid only for SOD, since for CAT, statistical difference was shown only comparing CON versus LPS group. Still, LPS versus LPS+25OHD3 no was statistically different (superscript b versus ab). Please, rewrite this sentence.
On item 3.3. (The last paragraph) The same recommendation as the previous item. Take care to consider statistical differences (or tendencies). In Table 4, is shown that no statistical difference between LPS and LPS+25OHD3 for D-lactate and IL-1b. Based on that, shouldn’t affirm that 25OHD3 prevents the LPS effect. Same recommendation for item 3.6 (for TNF-a jejunal and ileal), the LPS versus LPS+25OHD3 treatments no were statistically different.
On item 3.7 was mentioned about a
2.5. Discussion section
No comments
2.6. Conclusions section
No comments
Author Response
Response to Reviewer 1 Comments
Dear Reviewer
Good day. Thank you very much for your kind consideration with our submitted article and offering us the further opportunity to submit the revised manuscript. The amendments are highlighted in red in the revised manuscript. We have revised our manuscript for language and grammar checked by a native English speaker working in our university. We do thanks to your critical evaluation to make the manuscript more effective for review process in Antioxidants Journal.
Thank you and best regards.
Dr. Xiangshu Piao
State Key Laboratory of Animal Nutrition
College of Animal Science and Technology
China Agricultural University
Comments to authors:
Specific comments:
Abstract section
Point 1: I recommended briefly describing the protocol of LPS challenge and response parameters collected before the results paragraph. That information is essential for a quick review of potential readers and helps to best understand the whole paper.
Response 1: Thanks for your available comment. In the revised manuscript, I have briefly described the protocol of LPS challenge and response parameters collected before the results paragraph. See Line 17-21.
Introduction section
Point 2: In the first paragraph, the authors justify that the “broilers are frequently challenging” to many factors mentioned in the environmental pollution. Maybe to be more accurate with this proposition, “environmental pollution” could be changed by “climatic changes” and “global warming” as a direct stressor (principally as heat stress) that induces oxidative damage. Using the term “environmental pollution” involves many factors that not necessarily induce a direct stress response in the animals.
Response 2: Thanks for your suggestion. According to your advice, I have modified this sentence in the revised manuscript. See Line 40-42.
Point 3: The last paragraph of the introduction section is mentioned that: "Several studies demonstrated that adding 25OHD3 to the diet improved antioxidant status, immune function…", and in the subsequent paragraph: “However, fewer studies were conducted to explore the effect of 25OHD3… under LPS challenge”. Both prepositions are a little contradictory since the authors justify the lack of studies because the 25OHD3 benefits were not explored using the LPS model. Also, the principal hypothesis of the study is to show that 25OHD3 has antioxidant, anti-inflammatory, immunological, and gut health properties. It is important to highlight that LPS was used as a model of health challenge (as a tool) and does not necessarily represent a potential problem for broiler chick production like bacterial, fungal, or viral infections released in inflammatory diseases.
Response 3: Thanks for your advice. Indeed, both prepositions are a little contradictory in the last paragraph of the introduction section. I have modified this according to your comment. Moreover, I have added that LPS was used as a model of health challenge (as a tool) and does not necessarily represent a potential problem for broiler chick production like bacterial, fungal, or viral infections released in inflammatory diseases. See Line 63-76.
2.3. Materials and methods section
Point 4: The term “ad libitum” is repeated. Please, edit this paragraph.
Response 4: Thanks for your advice. I have corrected this issue. See Line 86-87.
Results section
Point 5: On item 3.1., I recommend expressing the results of body weight variation in the proportion of initial body weight (before inoculation) or declaring the average of initial body weight between treatments to elucidate that there is no difference at 21-d-old as a result of the treatments (specifically to 25OHD3 supplementation).
Response 5: Thanks for your advice. As shown in Figure 1, I have modified the presentation of BW loss results, and elucidate that there is no difference for BW before LPS challenge at 21-d-old among the treatments. See Line 178-181.
Point 6: On item 3.2. the authors mentioned that “25OHD3 supplementation prevented the LPS-induced decrease in the activities of CAT and SOD”. Also, this affirmation is valid only for SOD, since for CAT, statistical difference was shown only comparing CON versus LPS group. Still, LPS versus LPS+25OHD3 no was statistically different (superscript b versus ab). Please, rewrite this sentence.
Response 6: Thanks for your advice. I have corrected this issue in the Result and Abstract. See Line 188-189 and 23-24.
Point 7: On item 3.3. (The last paragraph) The same recommendation as the previous item. Take care to consider statistical differences (or tendencies). In Table 4, is shown that no statistical difference between LPS and LPS+25OHD3 for D-lactate and IL-1b. Based on that, shouldn’t affirm that 25OHD3 prevents the LPS effect. Same recommendation for item 3.6 (for TNF-a jejunal and ileal), the LPS versus LPS+25OHD3 treatments no were statistically different.
Response 7: Thanks for your advice. I have corrected these issues in the Result and Abstract. See Line 206-207, 232-233 and 25-28.
Point 8: On item 3.7 was mentioned about a
Response 8: Thanks for your advice. I have checked the description on item 3.7. And I have corrected the issue about fermentation metabolites of cecal digesta on 3.8. that is similar to the previous item.
2.5. Discussion section
Point 9: No comments
Response 9: Thanks for your advice.
2.6. Conclusions section
Point 10: No comments
Response 10: Thanks for your advice.
Reviewer 2 Report
The authors have evaluated the 25-hydroxycholecalciferol (25-OHD3) supplementation effects on antioxidant status, immune function, intestinal morphology and microbiota composition in models of piglets and chickens, and they have published a series of papers in the journals in the field of animal science [Reference No.15,16,17,19]. In this study, the authors established a chicken inflammation model by using LPS challenge, and then assessed the effects of 25-OHD3 on antioxidant capacity, immune status, gut barrier function, and microbiota composition of broilers under the LPS challenge. In my opinion, this article is not innovative enough. Moreover, the main insight is not focused and the lack of cellular and molecular mechanism results in this paper does not reach the publication standard of the journal, antioxidants.
Author Response
Response to Reviewer 2 Comments
Dear Reviewer
Good day. Thank you very much for your advice. The amendments are highlighted in red in the revised manuscript according to the comments of Reviewer 1 and Academic Editor. We have revised our manuscript for language and grammar checked by a native English speaker working in our university. We do thanks to your critical evaluation to make the manuscript more effective for review process in Antioxidants Journal. I hope you can take this revised manuscript into consideration.
Many thanks.
Sincerely yours.
Dr. Xiangshu Piao
State Key Laboratory of Animal Nutrition
College of Animal Science and Technology
China Agricultural University
Comments: The authors have evaluated the 25-hydroxycholecalciferol (25OHD3) supplementation effects on antioxidant status, immune function, intestinal morphology and microbiota composition in models of piglets and chickens, and they have published a series of papers in the journals in the field of animal science [Reference No.15,16,17,19]. In this study, the authors established a chicken inflammation model by using LPS challenge, and then assessed the effects of 25OHD3 on antioxidant capacity, immune status, gut barrier function, and microbiota composition of broilers under the LPS challenge. In my opinion, this article is not innovative enough. Moreover, the main insight is not focused and the lack of cellular and molecular mechanism results in this paper does not reach the publication standard of the journal, antioxidants.
Response: Indeed, we evaluated the effects of 25OHD3 supplementation on antioxidant status, immune function and intestinal health in piglets and broiler. However, few study have detected the effects of 25OHD3 supplementation on broilers under the LPS model. Although Morris et al. (2014) evaluated the effects of 25OHD3 supplementation on broiler during an experimental LPS injection, they just focused on inflammation response in the liver of broilers. In our study, the effects of 25OHD3 on broilers were specifically investigated in terms of antioxidant, intestinal barrier function and intestinal microbiota. In modern intensive production, broilers are exposed to frequent challenges such as pathogenic infection, drug abuse and other external elements, which could induce oxidative damage and immunological stress. Our results of this study can provide a theoretical basis for the application of 25OHD3 in poultry production under stress response. I sincerely hope that you can consider this revised manuscript. Thank you very much, and we will continue to work hard to correct our shortcomings.
Reference:
Morris, A.; Shanmugasundaram, R.; Lilburn, M.S.; Selvaraj, R.K. 25-Hydroxycholecalciferol supplementation improves growth performance and decreases inflammation during an experimental lipopolysaccharide injection. Poult. Sci. 2014, 93, 1951-1956.
Round 2
Reviewer 2 Report
What I concerned is that were these commercial ELISA KITS (IL-1β, IL-6, TNF-α, IFN-γ, IgA, IgG and IgM) are specific and available for determination poultry samples? Please provide literatures to support availability of these kits used in your study. Moreover, if they are available, what are the test lines for these kits? What is the coefficient of variation (intra-assay CV and/or inter-assay CV) of your data?
Author Response
Response to Reviewer 2 Comments
Dear Reviewer
Thank you very much for your kind consideration of our revised manuscript and for offering us the further opportunity to submit this manuscript. The amendments are highlighted in red in the revised manuscript. We do thank you for your critical evaluation to make the manuscript more effective for the review process in Antioxidants Journal.
Many thanks.
Sincerely yours.
Dr. Xiangshu Piao
State Key Laboratory of Animal Nutrition
College of Animal Science and Technology
China Agricultural University
Point 1: What I concerned is that were these commercial ELISA KITS (IL-1β, IL-6, TNF-α, IFN-γ, IgA, IgG and IgM) are specific and available for determination poultry samples? Please provide literatures to support availability of these kits used in your study.
Response 1: Thanks for your advice. I am sure that these commercial ELISA kits (IL-1β, IL-6, TNF-α, IFN-γ, IgA, IgG and IgM) are available for determining poultry samples. For detecting IgA, IgG and IgM, I also used assay kits from Nanjing Jiancheng Bioengineering Institute (Nanjing, China). These kits were purchased only through a Leadman agent, I have modified this in the revised manuscript. See Line 115-117. The literature was as follows:
Reference:
Wang, J.; Liu, S.; Ma, J.; Piao, X. Changes in growth performance and ileal microbiota composition by xylanase supplementation in broilers fed wheat-based diets. Front. Microbiol. 2021, 12, 706396.
Ma, J.; Mahfuz, S.; Wang, J.; Piao, X. Effect of dietary supplementation with mixed organic acids on immune function, antioxidative characteristics, digestive enzymes activity, and intestinal health in broiler chickens. Front. Nutr. 2021, 8, 673316.
Point 2: Moreover, if they are available, what are the test lines for these kits? What is the coefficient of variation (intra-assay CV and/or inter-assay CV) of your data?
Response 2: Thanks for your advice. The test lines for these kits, such as IL-1β, IL-6, TNF-α and IFN-γ are 0.5-200 pg/mL, 2-200 ng/L, 2-200 ng/L and 0.5-200 pg/mL, respectively. Our all results matched the test line. The values of inter-assay CV for L-1β, IL-6, TNF-α, IFN-γ, IgA, IgG and IgM in our study were 10.80%, 15.23%, 12.40%, 8.79%, 10.11%, 15.29% and 11.94%, respectively.